**Risk Factors and Perceived Restoration in a Town Destroyed by the 2010 Chile Tsunami**

Carolina Martínez[1, 6], Octavio Rojas[2], Paula Villagra[3], Rafael Aránguiz[4, 6], Katia Sáez-Carrillo[5]

[1]Institute of Geography, Pontificia Universidad Católica de Chile, Santiago, 8320000, Chile

[2]Department of Territorial Planning, Universidad de Concepción, Concepción, 4030000, Chile

[3]Institute of Environmental and Evolutionary Sciences, Universidad Austral de Chile, Valdivia, 5090000, Chile

[4]Department of Civil Engineering, Universidad Católica de la Santísima Concepción, Concepción, 4030000, Chile

[5]Department of Statistics, Universidad de Concepción, Concepción, 4030000, Chile

[6]National Research Center for Integrated Natural Disaster Management (CIGIDEN), Santiago, 8320000, Chile

Correspondence to: Carolina Martínez (camartinezr@uc.cl)

**ABSTRACT**

A large earthquake and tsunami took place in February 2010, affecting a significant part of the Chilean coast (Maule earthquake, Mw = 8.8). Dichato (37° S), a small town located on Coliumo Bay, was one of the most devastated coastal areas and is currently under reconstruction. Therefore, the objective of this research is to analyze the risk factors that explain the disaster in 2010, as well as perceived restoration six years after the event. Numerical modeling of the 2010 Chile tsunami with four nested grids was applied to estimate the hazard. Physical, socio-economic and educational dimensions of vulnerability were analyzed for pre- and post-disaster conditions. A perceived restoration study was performed to assess the effects of reconstruction on the community. It was focused on exploring the capacity of newly reconstructed neighborhoods to provide restorative experiences in case of disaster. The study was undertaken using the Perceived Restorative Scale.

The vulnerability variables that best explained the extent of the disaster were housing conditions, low household incomes and limited knowledge about tsunami events, which conditioned inadequate reactions to the emergency. These variables still constitute the same risks as a result of the reconstruction process, establishing that the occurrence of a similar event would result in a similar degree of devastation. For post-earthquake conditions, it was determined that all neighborhoods have the potential to be restorative environments soon after a tsunami. However, some neighborhoods are still located in areas devastated by the 2010 tsunami and again present high vulnerability to future tsunamis.

**Keywords**: tsunami, natural risk, territorial planning, social resilience

## 1. Introduction

A tsunami is a phenomenon known for its great destructive power in a short period of time; however, the process of post-disaster reconstruction usually lasts a long time and generates significant socio-territorial transformations. A total of seven destructive tsunamis affected the coasts of Indonesia, Samoa, Chile and Japan in the last decade alone: 2006, 2007, 2009, 2010 (Feb 27th and Oct 24th) and 2011. These tsunamis took the lives of 237,981 people and generated an estimated US $456 million in economic losses (Løvholt et al., 2012; Løvholt et al., 2014). These degrees of destruction have been explained by a number of factors, such as ineffective early warning systems, inadequate information management by the population, lack of emergency mechanism coordination and high levels of social vulnerability (Rofi et al., 2006; Løvholt et al., 2014). Although scientific research has led to significant advances in the understanding of the generation and propagation mechanisms of these phenomena (Aránguiz et al., 2013; Løvholt et al., 2014), other aspects linked to social components (vulnerability and resilience) are less understood, primarily for post-disaster conditions, given social system dynamics and complexity. The latest events have shown that increased mortality may be associated with intrinsic aspects of vulnerability, which in the natural disaster context is defined as the inability of society to respond to an event, in this case a dangerous natural phenomenon (Anderson and Woodrow, 1989 in Cardona, 2001; Wilches-Chaux, 1993). Intrinsic aspects include population characteristics such as age and gender (Rofi et al., 2006), income levels and occupations (Birkmann et al., 2007), ideological and cultural factors, levels of knowledge and inadequate reactions to the emergency (Ruan and Hogben, 2007). Other studies, through a line of still incipient work, have established that elements associated with social capital and territorial identity foster social resilience, which would be an enabling framework for overcoming the negative effects of a disturbance (Pelling, 2003).

The 2010 Chile tsunami showed the high fragility of social and institutional systems in coastal areas, as significant destruction along 600 km of coastline was observed (Quezada et al., 2010; Fritz et al., 2011; Jaramillo et al., 2012; Sobarzo et al., 2012; Bahlburg and Spiske, 2012; Martinez et al., 2012). Historical records show that these phenomena are not sporadic in the country but rather highly recurrent, causing significant devastation (Lomnitz, 1970; Monge, 1993, Lagos, 2000; Ruegg et al., 2011; Palacios, 2012). Territorial planning in Chile, as in much of the rest of the world, has been focused primarily on interventions for mitigation (Herrmann, 2015), with policies and instruments for reconstruction (e.g., Sustainable Reconstruction Plans and Master Plans) focused on housing production rather than social reconstruction of territories (Rasse and Letelier, 2013; Martinez, 2014). Thus, interdisciplinary approaches necessary for the reconstruction of human settlements in an integrated manner, i.e., studies which identify, assess and integrate physical, economic, social, environmental and perceptual factors, have been neglected. This complex approach has already been addressed in an international context, with the application of different study models of urban resilience to disaster (e.g., Cutter et al., 2008; Norris et al., 2008). Resilience refers to the ability of a community to adapt and recover after a disturbance without losing its character (Cutter et al., 2014; Walker and Salt, 2006), and is expressed multi-dimensionally (Cutter et al., 2014); in Chile, however, its psychological and social dimensions are the least considered in post-disaster planning. This is the case despite the fact that the integration of these dimensions in planning can promote community recovery after a disaster, with the potential to rebuild "the place where the restoration occurs" (Allan and Bryant, 2010). A restorative experience is described as "the process of recovering psychological and social resources that have become diminished in the efforts to meet the demands of everyday life" (Hartig, 2007, p.164). After a large tsunami, the city

"takes on a new meaning [and] its spaces and components are re-evaluated (by the people)" for their
capacity to provide restorative experiences (Allan and Bryant, 2010). Thus, post-disaster reconstruction
processes are an opportunity to effectively reduce risk and generate mechanisms of physical as well as
social resilience.
In this context, we analyze pre- and post-disaster tsunami inundation risks in one of the coastal towns
most affected by the earthquake and tsunami on Feb. 27, 2010, which presented an intense
transformation as a result of post-disaster reconstruction. It is unknown whether this reconstruction
process has reduced vulnerability and provided a restorative urban system, enhancing urban resilience,
or if it has generated new risk areas. Questions were asked regarding the neighborhoods being rebuilt in
Dichato such as: Do they have the potential to be restorative environments? Which specific sites provide
restoration? Are restorative environments pre-existing areas that persist after the disaster, or are they
new sites built during reconstruction? These questions seek to determine whether the reconstruction
process has promoted the population's ability to adapt after a tsunami, and whether reconstruction
decisions have decreased the potential for damage in the case of future events.
**2.    Regional setting**
Dichato is a town located on Coliumo Bay (36° 33'S). It belongs to the Tomé Commune and has a
population of 3,488 inhabitants dedicated largely to fishing, trade and tourism (INE, 2002).
Dichato's urban layout is characterized by a coastal plain of approximately 2 km$^2$, dissected by Dichato
Stream, with an average height of 6 m (Fig. 1). These characteristics explain the great impact of the
2010 tsunami, which had inundation heights of up to 8 m, a penetration distance of 1.3 km inland and an
inundation area of 0.85 km$^2$. The affected population was 1.817 people, with 66 dead and 60% of all
housing destroyed (Martinez et al., 2011). According to historical records, this coast had previously been
affected by six destructive tsunamis, the most significant occurring in 1751 (M = 8.5), 1835 (M = 8.2)
and 1960 (M = 9.5) (Lagos, 2000; Palacios, 2012).
**3.    Materials and methods**
In order to give risk a value in pre- and post-disaster conditions, the equation R = H * V was used, where
R = Risk, H = Hazard and V = Vulnerability (Blakie et al., 1994).
3.1 Hazard
The tsunami hazard was estimated by means of a numerical simulation considering the tsunami on
February 27, 2010. The Non-hydrostatic Evolution of Ocean WAVEs (NEOWAVE) numerical model
(Yamazaki et al., 2009, 2011) was used. This model solves linear and nonlinear shallow water equations
using nested grids with different spatial resolutions. In this case, 4 nested grids were used, with
resolutions of 120" (~3600 m), 30" (~900 m), 6" (~180 m) and 1" (~30 m). Grids 1 and 2 were built
from GEBCO topo-bathymetric data, while nautical charts and detailed bathymetry in Coliumo Bay
were used for Grids 3 and 4. In addition, 2.5-m-resolution LIDAR topographic data obtained in 2009
were used for Grid 4, representing the situation at the time of the 2010 tsunami. The initial tsunami
condition was defined using the finite fault model proposed by Hayes (2010), with 180 sub-faults and
heterogeneous slip. Figure 2 shows the 4 nested grids and the tsunami initial conditions used in the
numerical simulation. The figure shows that Grid 4 takes into account the entire Coliumo Bay and not
just the town of Dichato.

A Manning roughness coefficient of 0.025 was used and the total simulation time was 6 hours, with
output results (water surface elevation and current velocity) of 1 minute. The tide level was set to the sea
level at the time of the maximum inundation. To do this, preliminary numerical simulations were
conducted to find the maximum tsunami wave. The tide level was estimated to be -0.25 m, which the
grids were modified to include. Furthermore, a virtual tide gauge on the Dichato beachfront was defined
to obtain arrival times of different tsunami waves. The validation of the numerical simulation was
performed using the Root Mean Square Error and the parameters $K$ and $\kappa$ proposed by Aida (1978), cited
by Suppasri et al. (2011) and given below in equations 1 and 2. The variable $K_i$ is defined as $K_i = x_i/y_i$,
where $x_i$ and $y_i$ are recorded and computed tsunami heights, respectively. The recorded tsunami heights
were obtained from field survey data published by Mikami et al. (2011) and Fritz et al. (2011).

$$\text{Eq (1)} \qquad \log K = \frac{1}{n}\sum_{i=1}^{n} \log K_i$$
$$\text{Eq (2)} \qquad \log \kappa = \sqrt{\frac{1}{n}\sum_{i=1}^{n}(\log K_i)^2 - (\log K)^2}$$

Hazard levels proposed by Walsh et al. (2005), defining flow depths of 0, 0.5 and 2.0 m, were selected
when obtaining tsunami inundation hazard levels (Table 1). The hazard levels generated by the current
velocity were also included in the hazard analysis. The levels were selected in terms of security for
human life (Table 2).

3.2 Vulnerability

The overall vulnerability analysis was performed for two scenarios: pre- and post-disaster. This was
done in order to establish the variables related to the level of destruction and the effects generated by the
process of post-disaster reconstruction in the configuration of new risk areas. Pre-disaster vulnerability
was determined in 2010, while post-disaster conditions were determined using data collected in 2014.

In the case of pre-disaster conditions, the analysis unit corresponded to census blocks. Due to the
destruction of most of the census blocks after the tsunami, for post-disaster conditions the analysis unit
was the neighborhood. Neighborhood units were first defined with respect to their pre- and post-tsunami
existence. In this line, newly reconstructed areas and old built zones were identified. Within each of
these larger areas, neighborhoods were subdivided based on urban morphology characteristics such as
type of grid, housing and land use. Finally, main streets such as avenues were used to define the
boundaries of neighborhoods. Fig. 1 illustrates the location of the nine neighborhoods in Dichato.

For the overall vulnerability analysis, variables selected for both scenarios were representative of
physical, socio-economic and educational dimensions; however, some variables were modified
according to the pre- and post-disaster availability of data (Table 3). For pre-disaster conditions, the
variables related to the physical and socio-economic dimensions were obtained from the 2002 census
(INE, 2002). However, due to the destruction of the post-tsunami census blocks, the use of the census
was invalidated and data were collected through surveys. To this end, variables related to those of the
pre-disaster data collection were selected (for example, in the physical dimension the housing type
variable is related to the number of floors variable). The survey was also used also to collect data related

to the educational dimension in pre- and post-disaster scenarios, including the following variables: knowledge of evacuation routes, knowledge of safe zones and participation in educational programs or talks.

Three levels of vulnerability for each dimension were established (high, medium, low). The variables associated with each vulnerability dimension were given equal weight in the final matrix. This was done because previous studies that included the dimension of vulnerability in risk (Martínez et al., (2012); Rojas et al., (2014) used similar criteria, which have been found to be representative of local conditions. Variables were incorporated into the GIS using ArcGis 10.1 to generate thematic maps and summary charts through map algebra.

3.3 Environmental restoration

The capacity of the neighborhoods of Dichato to provide restorative post-disaster experiences was assessed through a perceived restoration study (Hartig et al., 1997). This type of study explores the capacity of environments to be restorative, which is particularly crucial in cities prone to natural disasters such as tsunamis. For Dichato in particular, this type of study can reveal the specific landscape attributes that influence perceptions, indicating whether restorative post-tsunami experiences are based on newly reconstructed neighborhoods and features or pre-existing neighborhood character. In other ways, it provides information regarding whether or not post-tsunami reconstruction contributes to a restorative experience, and hence to resilience.

To undertake this study, the nine neighborhoods of Dichato were defined as analysis units (Fig. 1). People living in each of these neighborhoods assessed their environment by means of the Perceived Restorative Scale (PRS), an instrument constructed based on the Attention Restoration Theory (Kaplan and Kaplan, 1989). The PRS has been used to identify landscape attributes that can be restorative to people subjected to high levels of stress and mental fatigue (Hartig et al., 1997; Korpela and Hartig, 1996; Ulrich et al., 1991). Three factors of the PRS were used to evaluate the interaction of people with the neighborhood they inhabit: being away (BE-AW), which reflects the need to escape from everyday life or daily mental activities that require major concentration; fascination (FAS), which is found in environments that attract and hold our attention without any effort; and compatibility (COMP), which refers to a sense of oneness with environments that provides the capability to meet our desires and needs. A survey was developed based on previously validated PRS questionnaires, including 15 sentences that refer to each of these factors (five for each). This survey was used to interview people in their homes during the same time the vulnerability questionnaire (described in the previous section) was administered. Each interviewee was informed about the survey takers' interest in understanding their experience in their current neighborhood. Accordingly, they were asked to assess each sentence from 1 to 7 (Likert scale), with 1 indicating that the sentence did not apply to their neighborhood at all and 7 indicating that it applied to the highest degree. As an example, a sentence referring to the BE-AW factor was "Being in this neighborhood is like a break from my daily routine." Another sentence, referring to the FAS factor, was "There are many interesting things that catch my attention in this neighborhood." A sentence referring to the COMP factor was "The activities that can be done in this neighborhood are activities that I enjoy." After evaluating each of the 15 sentences, each person was asked to describe the neighborhood areas they recalled during the evaluation.

Numerical data obtained using the Likert scale were organized in Excel sheets for each interviewee and
neighborhood, and were later used for performing an ANOVA and calculating mean and SD values for
each factor and neighborhood. Textual data were organized in Excel sheets as well, including the
frequency of mention of each site that interviewees recalled during the evaluations (expressed as a
percentage). The results allowed the following aspects to be identified: significant differences among
neighborhoods with respect to restoration capacity; neighborhoods with the highest and lowest
restoration values and the sites that were more meaningful for providing a restorative experience to the
inhabitants.
3.4 Sampling and statistical analysis
For the application of pre-disaster surveys oriented at determining vulnerability to and perception of the
phenomenon, stratified sampling was conducted, with groups (strata) corresponding to 95 census blocks
(Figure 1) (INE, 2002). Population was defined as the number of inhabitants older than 15 years of age
$(N = 2120)$, with a confidence level of 95% and a sampling error margin lower than 5%. Finally, 337
surveys (n) were carried out.
The determination of post-disaster vulnerability and restoration was also addressed by stratified
sampling, where groups (strata) corresponded to 9 neighborhoods (Figure 3). Population was defined as
heads of households (male or female) who live in the town of Dichato permanently $(N = 1850)$. Eq (3),
for finite populations, was applied to determine the sample size.
Eq (3) $$n \geq \frac{N z_{1-a/2}^2 PQ}{z_{1-a/2}^2 PQ + d^2 (N-1)}$$
Where: confidence level was 95%, precision (5%) and proportion 90% ($\approx$ 90% of families in the Biobio
Region who experienced problems due to the 2010 earthquake and tsunami) (Larrañaga and Herrera,
2010). The minimum sampling size was estimated to be n=130. Finally, 156 surveys were carried out.
Data collected in these surveys were subjected to multivariate descriptive analysis, cluster analysis and
principal component analysis. Cluster analysis allowed the similarities and dissimilarities among
neighborhoods to be easily observed, whereas principal component analysis allowed the association of
the different dimensions of vulnerability with each analyzed neighborhood to be observed. The chi-
squared test was used to compare proportions and a one-way analysis of variance (ANOVA) was
conducted for the numerical variables. The Tukey test was applied for comparison, using a significance
level of $\alpha = 0.05$.
3.5 Risk
Risk factors were integrated into a matrix (Eckert et al., 2012; Jelínek et al., 2012; Martínez et al., 2012)
and three risk levels were obtained from the multiplication: high, medium and low, with scores from 1 to
9 (Table 4). Risk level is applied to analysis units, according to pre- and post-event conditions, in the
GIS vulnerability section.
**4.     Results**
4.1     Hazard
Fig. 3 (a) shows the inundation area obtained from the numerical simulation. Dots indicate inundation
height measurements from field surveys (Mikami et al 2011; Fritz et al, 2011), while the asterisk
indicates the synthetic tide gauge location. Fig. 3 (b) shows a comparison of recorded and simulated
data, where the error obtained from Eq (1) was K = 1.09 with a standard deviation from Eq (2) of $\kappa$ =
0.12, which is considered acceptable (Suppasri et al., 2011). Fig. 3 (c) shows the tsunami wave form
obtained from the synthetic tide gauge. It can be seen that the largest wave is not the first, but rather the
third wave, which reached an inundation height of up to 7m. A fourth wave is also observed, reaching up
to 5m.  Fig. 4 shows the area inundated by the event, which reached a maximum runup of 10m, spread
through Dichato Stream.
4.2     Pre-disaster vulnerability
In the case of physical vulnerability, 51% of census blocks reported high vulnerability levels, which
involved 47% of the total inundated area and 57% of the total population (Fig. 5). Vulnerability levels
relate to a deficiency in housing materials (e.g.: wood, fiber cement, clay and waste). In relation to
economic vulnerability, 73% of households reported a medium level, which involves 61% of the
inundated area and 67% of the total population. These vulnerability levels can be explained mainly by
the location of the residential areas, in which more than 75% of the inhabitants reside, where there is no
overcrowding but income levels are low, with approximately 44% of the population receiving monthly
incomes of less than $118,000 Chilean pesos (about US $170). In addition, 54% of the population has
high educational vulnerability. Low levels of schooling influence overall vulnerability as 42% of the
population has only a primary school education or has not completed this level and only 55% has
finished secondary school. It is important to note that 58% of the population indicated that the tsunami
was the result of the earthquake, while 42% attributed the tsunami to other causes, sometimes related to
religion, including global warming and the apocalypse.
4.3.    Post-disaster vulnerability
For post-disaster conditions, the Reconstruction Plan applied to Dichato, known as PRBC-18, modified
29% of the total town area, with 15% established as a tsunami-resistant building area, not including
expropriation (Fig.7). Elevated houses (stilt houses) and community buildings were designed and built in
these areas (coastline). Twelve percent of the total area was reserved for mitigation parks, which were
built along the coastline and river banks, where the tsunami surged and the greatest destruction was
generated. Construction of the mitigation park began in 2015, with a line of trees that covered a surface
area 95.7 ha, with a width of more than 20 m. The fishing area accounts for 1.6% of the total town area,
including the construction of a fishing pier and a market in Villarrica Cove.
The post-event vulnerability measures presented variations in relation to pre-disaster levels. The
physical dimension presented an average level in 53% of the total area, explained by the number of
floors ($\geq$2) and the type of dwelling. Regarding socio-economic vulnerability, the reported levels were
medium and high in 42% and 31% of the total area, respectively, and were associated with a low per
capita income level. Educational vulnerability was average in 87% of the area; however, spatial
differences were reported among the districts analyzed. The cluster analysis (Fig. 8a) performed for
post-event vulnerability dimensions identified six neighborhood groups. Four groups were represented
individually by neighborhoods C, E, F and A. The fifth cluster grouped analysis units D and B. Finally,
the sixth group was composed of units I, H and G. Only neighborhoods C, E, A, D and B were directly
affected by the 2010 tsunami inundation.
The ANOVA showed significant differences in the physical and educational vulnerability dimensions ($p$
<0.05), while the socio-economic dimension was homogeneous for all evaluated neighborhoods ($p$ =
0.1808). The neighborhoods with the highest physical vulnerability were older sectors (I, D) and a
provisionally relocated sector (A). Neighborhoods directly affected by the tsunami (B, C) were grouped
in the medium level, as was as an unaffected sector (H). The neighborhoods found in the low level (E, F,
G), presented higher quality buildings. Regarding the educational dimension, the lowest vulnerability
corresponded to relocated sector A, which was most devastated by the 2010 tsunami. The above was
reinforced by a principal component (PC) analysis, which showed that the first two components
explained 85.5% of the total variance. Fig. 8b indicates that only sector C had a higher association with
socio-economic vulnerability, while the remaining 8 neighborhoods were related to the physical and
educational vulnerability dimensions.

Regarding feelings assessed regarding the possibility of a future tsunami (Table 4), 5 feelings showed no
significant differences by neighborhood ($p$>0.05): panic (19%), fear (39%), tranquility (41%), security
(19%) and indifference (3%). A significant difference ($p$ = 0.0258) was found for the feeling of anxiety,
which was higher (67%) for relocated inhabitants (A).

The overall vulnerability analysis for pre- and post-tsunami conditions indicates that vulnerability was
reduced from high to medium levels, and that the spatial distribution of vulnerable areas was maintained
for both conditions (Fig. 9). For pre-tsunami conditions, 90% of the area evaluated presented high
vulnerability, 5.4% medium vulnerability and 4.6% low vulnerability. For post-tsunami conditions, 55%
of the area presents high vulnerability levels and 45% exhibits medium vulnerability, while low
vulnerability was not found. These findings lead to the conclusion that the entire area has high or
medium levels of vulnerability.

4.4.    Post-disaster environmental restoration

Safety perception in current residential areas was evaluated as safe or very safe by 65% of the local
population ($p$>0.05) in view of changes made by authorities in the Master Plan for Reconstruction. The
feeling of identity with the city pre-disaster was 49% ($p$>0.05), with higher percentages in
neighborhoods A (75%), C (59%) and D (57%), which were the most affected. Desire to change one's
place of residence was not homogeneous among neighborhoods ($p$ = 0.0018) and percentages ≥ 40%
were obtained both in sectors affected by the tsunami (A, B) and those not directly affected (F, G).
The 1-7 Likert scale was applied to assess 5 topics, namely reconstruction, process quality, equipment
and the role of the National Emergency Office (ONEMI). The results showed no differences among
neighborhoods. Positive evaluations were obtained for the reconstruction process (Mean = 5.6; SD =
1.4), associated equipment (Mean = 5.5; SD = 1.4) and quality (Mean = 6.0; SD = 1.3). The worst scores
were obtained for ONEMI (Mean = 3.8; SD = 1.9).

In relation to the perceived restoration study, the ANOVA showed significant differences among
neighborhoods ($p$ <0.05). The best evaluated areas were neighborhoods C (Mean = 6.1; SD = 0.8) and I
(Mean = 5.8; SD = 1.0).  The means reported here correspond to the three factors combined for each

neighborhood. For the results of each factor, see Table 5. Neighborhood C (Villarrica) was affected by the tsunami and completely rebuilt, while neighborhood I (Pingueral) was not modified. The new coastal infrastructure (25%), new anti-tsunami houses (19%) and views of the coast (16%) were most mentioned by respondents in neighborhood C as elements that contribute to restoration. Meanwhile, in neighborhood I, views of the river (20%) and the presence of uphill streets (20%) and nearby hills (17%) were the most mentioned as restorative elements. In contrast, neighborhoods A (Mean = 4.4; SD = 1.6) and H (Mean = 4.7; SD = 1.5) were the worst evaluated. Neighborhood A, as previously mentioned, is the relocated neighborhood most affected by the tsunami. In this case, new urban infrastructure (19%), views of the bay (19%), and the community building (19%) were found to contribute the most to restoration. Neighborhood H was neither affected by the tsunami nor modified. In this case, the presence of nearby hills (32%), pre-existing housing (23%) and uphill streets (16%) were found to add to restorative experiences.

4. 4    Pre- and post-disaster risk

Little difference was presented between the surface area (0.11 km$^2$) and spatial distribution of the tsunami risk areas, considering the pre- and post-2010-event conditions, with a high level of risk (≥78%) for both scenarios (Fig. 10). In case of pre-disaster conditions, some sectors of the town had small areas with medium risk, explained by better building quality or sites without buildings. This situation changed post-disaster due to increased construction, especially public housing as part of the Reconstruction Plan. As seen in Fig. 11, most buildings were destroyed by the tsunami due to poor quality of materials. In the area where the greatest destruction occurred (the Villarrica sector or section C), initial housing was replaced by two-level, 27-m$^2$ stilt houses made of wood, on 2-m-high steel columns. This type of housing was implemented as a mitigating action against the possibility of a tsunami with similar characteristics, in which the steel structure would prevail while the wood could always be replaced (Fig. 11E). Not all former inhabitants returned to their neighborhood to occupy these homes as most were elderly and could not climb stairs to enter the stilt houses (Khew et al., 2015). Despite being owners, they opted for relocation to higher sectors of Dichato, forming a new neighborhood. These homes were also located in neighborhood B (or downtown), where a beachfront and boulevard have been built in order to promote tourism. The only structural changes made to houses consisted of replacing steel columns with reinforced concrete columns, while retaining the same overall dimensions (Fig. 11F). The ground floors of these stilt houses have been transformed by the inhabitants in order to increase living area (Khew et al., 2015).

Neighborhood B presented the greatest post-earthquake transformation, replacing a fish market with beachfront buildings, a mitigation park and a boulevard with a striking design in order to attract tourism. However, behind these buildings, a mixture of stilt houses and other types of one-story housing, made of wood or masonry, were built, giving rise to new neighborhoods and post-earthquake risk areas. Other areas, such as neighborhood A, went from being provisional neighborhoods to consolidated settlements (e.g., El Molino) and in turn absorbed part of the relocated population. In general, new post-disaster risk areas affect neighborhoods C, D, E, F, G and H up to a height of about 20 meters; however, Dichato Stream extends the propagation area into more inland neighborhoods.

**5.    Discussion**

The main results of this research were that in this urban area, with its strong reliance on natural resources (fishing) and associated tourism, high risk levels are presented for both pre- and post-disaster conditions. These conditions are not new and have already been reported in other areas in the country under the reconstruction process (Rojas et al., 2014). However, few studies exist worldwide on how coastal towns evolve in response to post-disaster reconstruction processes, generating transformations that do not contribute to risk reduction or urban resilience.

The main factors explaining high risk levels are building quality and materials, which are highly related to the degree of destruction caused by the 2010 tsunami. Many of these houses were one-story buildings made of wood or lightweight materials and built in a do-it-yourself manner. Lack of infill and reinforced concrete masonry (failure of brick masonry infill walls and lightly reinforced concrete columns) was a damage factor, coinciding with studies by Palermo et al. (2013) in the area. According to these authors, residential housing consisting of light timber frame and concrete frame construction with brick masonry infill walls suffered widespread damage throughout the surveyed coastal region of Chile.

According to numerical modeling, tsunami wave heights reached between 5 and 8 m, with current velocities greater than 2 m/s, which could be enough to damage house foundations and destroy coastal infrastructure. In this regard, tsunami fragility curves developed by Mas et al. (2012) for Dichato from field data and satellite imagery showed a 68% probability of damage at a flow depth of 2 m, mainly due to building materials, predominantly wood. In this case, it was found that approximately 80% of the built area of Dichato experienced damage or was completely destroyed by the 2010 tsunami.

Other important factors were the socio-economic status and education level of the population, which were relevant mainly in pre-disaster conditions. In this case, 44% of the population has low incomes and a widespread lack of knowledge concerning emergency plans or evacuation routes, resulting in inadequate reactions. One year after the event, people still had symptoms of post-traumatic stress, indicating feelings of panic, fear and sadness (Venegas, 2011). Most of this population, which consisted of owners affected by the tsunami, moved to provisional neighborhoods where they remained for two years without basic services. Some of these provisional houses became final settlements. The study conducted in Dichato by Shahinoor and Kausel (2013) stated that risk of tsunamis is not well addressed in planning and community-oriented programs and that the pre-established mechanisms for post-disaster recovery are not appropriate, which is why risk is not reduced. The latter is not a specific problem of the location but derives from the lack of coordination between planning instruments and risk management in Chile, which is essentially reactive and not preventative (Martínez, 2014). On the other hand, Chile lacks a public policy oriented at establishing criteria or a reconstruction model to implement in case of a disaster, and usually gives priority to physical reconstruction rather than social reconstruction. Yet physical reconstruction continues to take place in inappropriate locations and therefore, in considering only spatial location, risk areas fail to be managed in a way that effectively reduces risk.

A limitation in this research was that not all of the variables used for post-disaster conditions were similar to those used for pre-disaster conditions, for which there were census data at census block level. Even so, the results were concordant with the observed reality and the behavior of the variables coincides with other studies carried out in affected locations (Martínez, 2014, Rojas et al., 2014). The lack of standardized open-access geospatial databases prior to the disaster makes it difficult to manage risk, overcome the emergency and begin the reconstruction process. In the case of Chile, this has been

critical, limiting the monitoring of post-disaster socio-territorial changes as since 2002 there has been no
update to the national census, which only recently has been planned for 2017. In addition, the lack of
coordination among various ministries in the country causes replication of information, which in turn is
collected according to various criteria or standards, and to which access is not easy. These aspects limit
continuity in risk studies. Given the high social cost of these and other threats, debate on this matter has
recently begun in the country.
Regarding restoration results, it is interesting to find that the restoration capacity of neighborhoods
varies with respect to the presence and absence of natural and built elements. Natural elements such as
the presence of hills and views of bodies of water contribute to perceived restoration. These results are in
line with previous restoration studies indicating that the presence of natural elements such as water and
vegetation are related to restorative environments (Korpela and Hartig, 1996, Hartig et al., 1997).
However, in this case, it is not only the mere presence of these elements that is relevant, but most
probably the sense of security they give to the community as well. Hills are useful for refuge in case of
tsunami as well as for observation points, which are much needed to keep people informed about what is
happening in the event of a disaster. Consequently, it is important for future planning processes to
consider the potential of natural elements to remain in place after a disaster in order to restore
communities post-disaster. For instance, access to these natural sites from different neighborhoods
should be enhanced during the reconstruction process by, for example, including evacuation routes that
lead to these areas for recreation purposes in everyday life. The latter would contribute to post-disaster
adaptation and social resilience (Pelling, 2003).
In addition, the restoration results show that new elements introduced during the reconstruction process
such as the coastal infrastructure for mitigation and anti-tsunami housing are also characteristic of
neighborhoods that provide restorative experiences (Khew et al., 2015). It is possible that these
elements, although they are built features, give a certain sense of security to respondents, which could
explain these results. This study did not focus on establishing relationships between perceived safety and
post-disaster restoration factors; however, it is highly recommended that this possible relationship be
expanded in future studies. It may be that post-disaster restorative experiences are found in new built
and mitigation sites that give security to the community. This would also be important to consider in the
reconstruction process, as built features of the kind described here not only play a role in mitigation, but
also in post-disaster community functioning, contributing to social resilience (Pelling, 2003).
In this sense, vulnerability and resilience are distinct elements but superimposed in their role in natural
disasters and brought together in the cornerstone of sustainability (Turner, 2010). In the case of Dichato,
vulnerability exhibited a close relationship to lack of resilience because few lessons were learned from
the 2010 event and the same mistakes are still being made, with a nearly completed rebuilding process
that presents vulnerability conditions very similar to those that existed prior to the 2010 earthquake. This
situation is explained by the emphasis on physical rather than social reconstruction, the lack of public
policies to face a rebuilding process of this magnitude despite recurring events in the country and,
especially, by the poor consideration and assessment of risk areas in planning at a local scale, since other
affected areas were repopulated in the same manner and relocated to the same risk areas (Martinez,
2014). In some neighborhoods, increased social and environmental problems such as pollution, crime
and poverty occurred as a result of reconstruction processes (Rojas et al., 2014). The main disadvantage
of these programs is that they were implemented as similar projects in 18 affected coastal towns,

regardless of geographic reality and territorial identity. In addition, the programs did not distinguish

between rural or urban areas. Small fishermen's coves located in coastal wetlands and small bays that

underwent semi-urbanization processes had to absorb relocated populations from affected areas,

resulting in increased population densities in new risk areas and loss of cultural and territorial identity.

The latter was reflected in that between 57% and 75% of the population most affected by the tsunami six

years ago identified with Dichato prior to the tsunami. In this respect, most current approaches establish

that resilience is characterized by socio-ecological system responses to natural disturbances, capacity for

self-organization, learning and adaptation to change (Folke, 2006; Turner, 2010). These elements present

a challenge from an institutional point of view in Chile, as risk management and its link to organized

society must be strengthened to ensure that investments in reconstruction processes effectively reduce

risks associated with phenomena that will undoubtedly continue to occur in the country. In addition,

reconstruction involves addressing physical, social and environmental territory components to facilitate

the development of post-disaster resilience, meaning that the country must change its approach to natural

disaster management, moving towards sustainability of its cities and coastal towns.

## 6.      Conclusions

The vulnerability factors that best explained the extent of the 2010 tsunami disaster were housing

materials, low incomes and poor knowledge about such phenomena, which conditioned inadequate

reactions at the time of emergency. The current town configuration, resulting from reconstruction

process in the six years after the event, has generated new risk areas that occupy the same locations as in

pre-event conditions, meaning that risk is not reduced. For post-earthquake conditions, it was determined

that all neighborhoods have the potential to be restorative environments after a tsunami, but with

different intensities, depending on the type of natural and built features they have kept and incorporated

during the reconstruction process. However, risk analysis indicates that neighborhoods with greater post-

disaster restoration ability remain in the same areas devastated by the 2010 tsunami and will likely be

destroyed again in a future event, a situation that forces us to reflect on how to plan coastal zone land

use and manage risk in the country. Finally, the methods used in this investigation, in which cluster

analysis, principal component analysis and a GIS platform are integrated, prove highly beneficial in an

interdisciplinary approach such as this, where the selection criterion of the variables is fitted to the

knowledge of the place and the variables are validated as a function of the local reality, which facilitates

their adaptation to other geographical areas of interest.

**Acknowledgements**

Grants funding from FONDECYT N°1151367, 11140424, 1150137 and FONDAP N°15110017 from

the National Scientific and Technological Research Commission (CONICYT) of Chile and from DID I-

2014-11 from the Universidad Austral de Chile. 3. The authors thank the referees (A. Armigliato) for the

comments on the manuscript.

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

Fig. 1 Geographical context of the study area. Dichato is located on Coliumo Bay (36° S). The letters
(A-I) identify different neighborhoods analyzed in the post-disaster period.

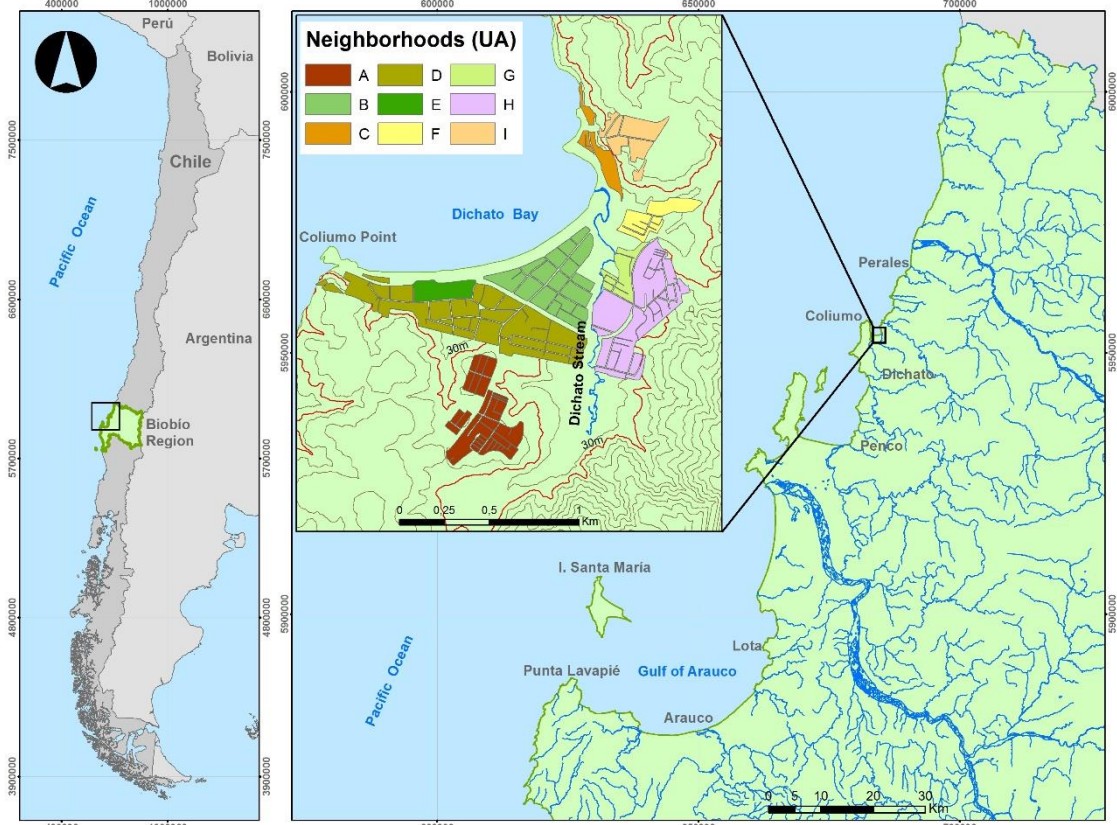


Fig. 2. Nested computational grids. The inset in Grid 2 shows the tsunami initial condition (Hayes,
2010). TG in grid 4 indicates the location of the virtual tide gauge.

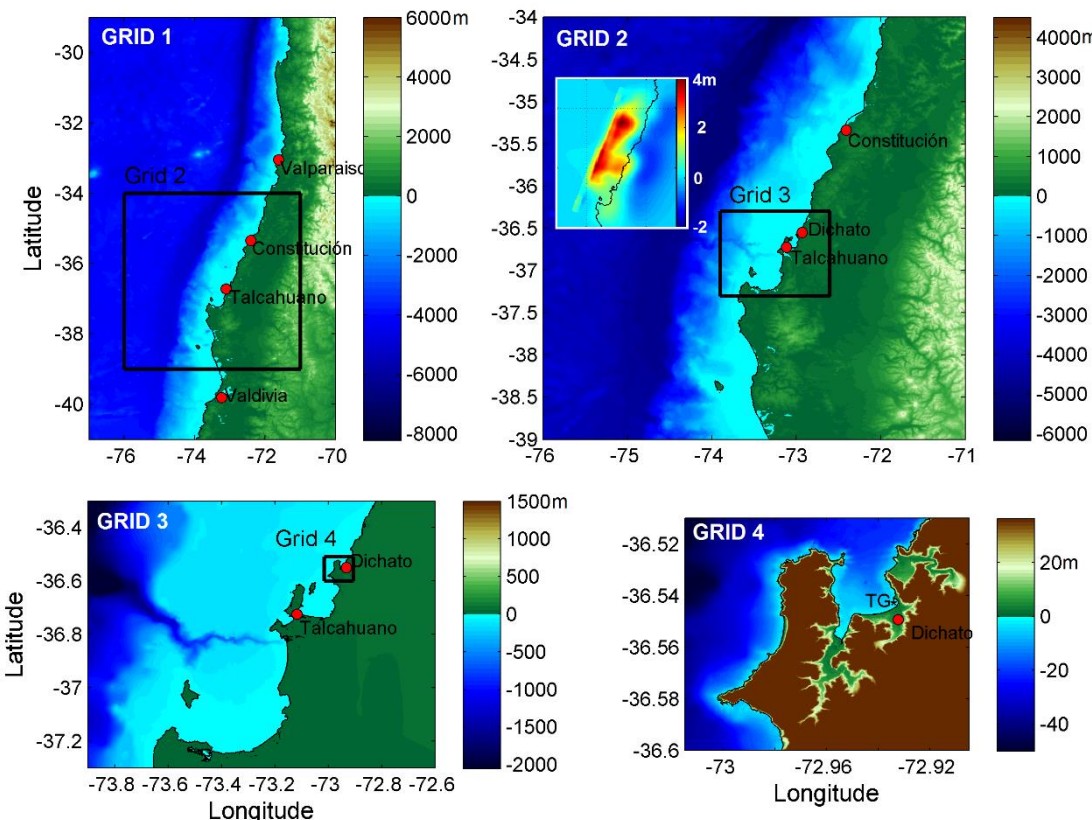


Fig. 3. (a) Inundation area obtained in the numerical simulation. (B) Comparison of measured and simulated data. (C) Tide gauge on Dichato beachfront. The solid line represents the measurement equal to the simulation, while the dotted lines show the measurement +- 2 m. The simulation shows results in a range of +- 2 m with respect to the measured data.

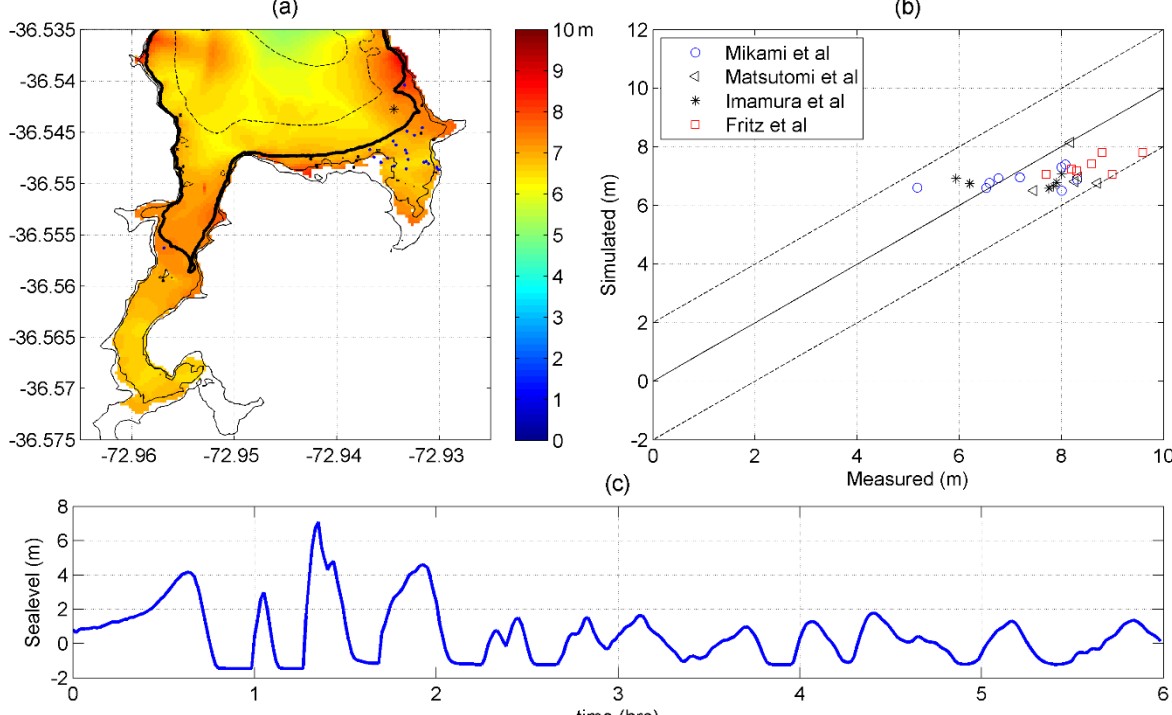

Fig. 4. Tsunami hazard maps. A) Current Velocity. B) Flow depth

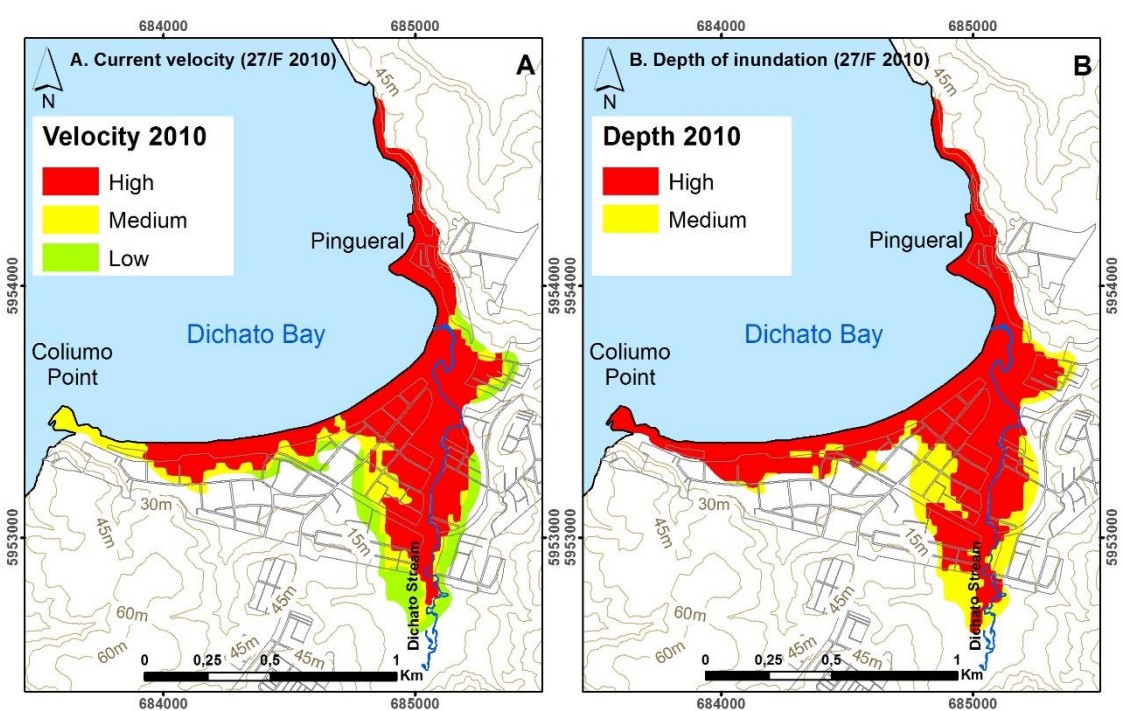

Fig. 5. Pre-event tsunami vulnerability of the town of Dichato

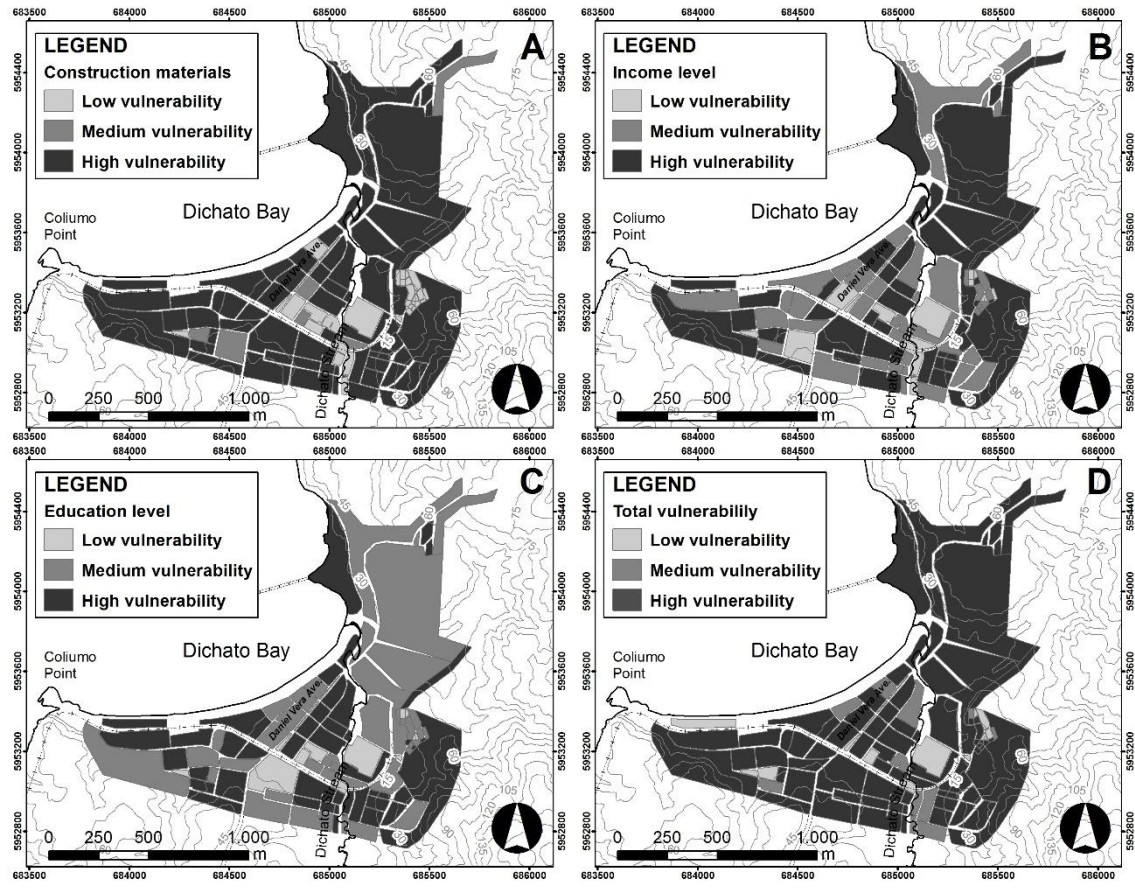

775                   Fig. 6. Post-event Tsunami vulnerability of the town of Dichato

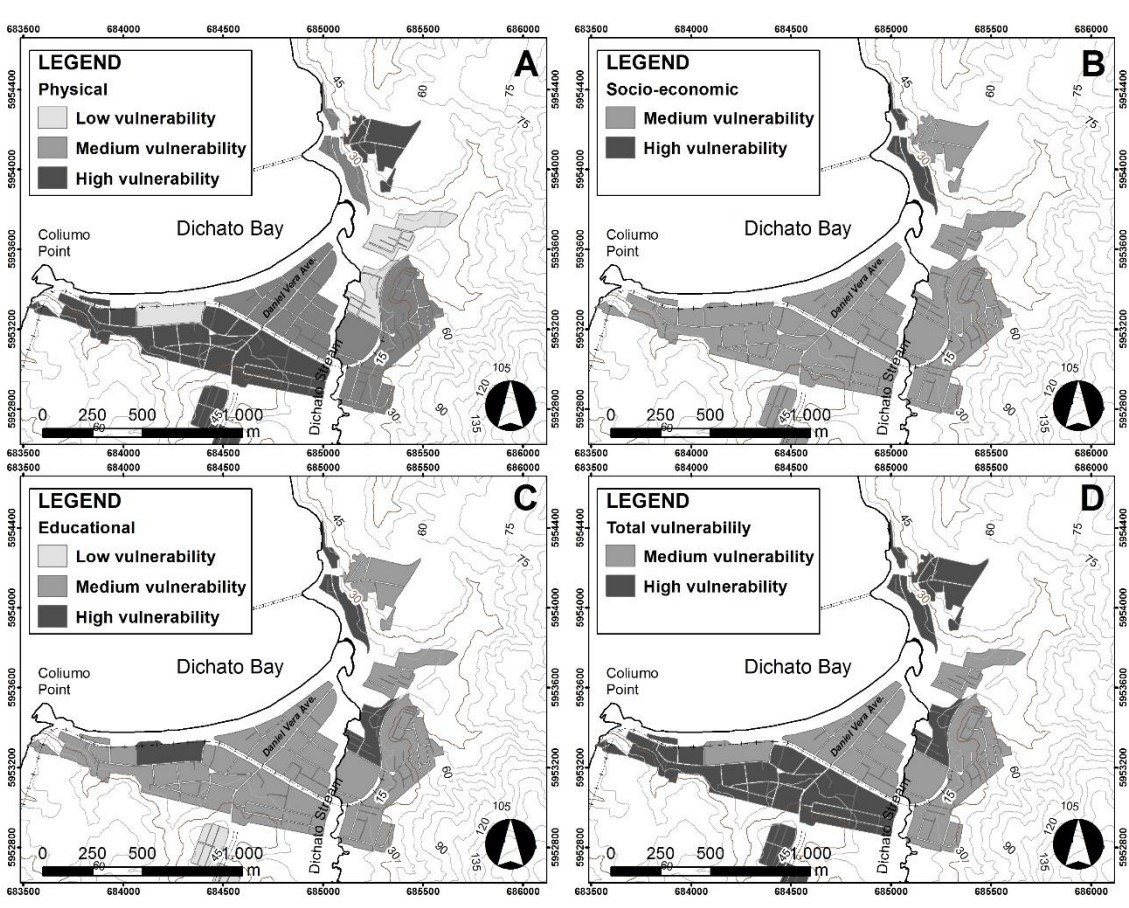

778                   Fig.7. Master Plan for Reconstruction for Dichato

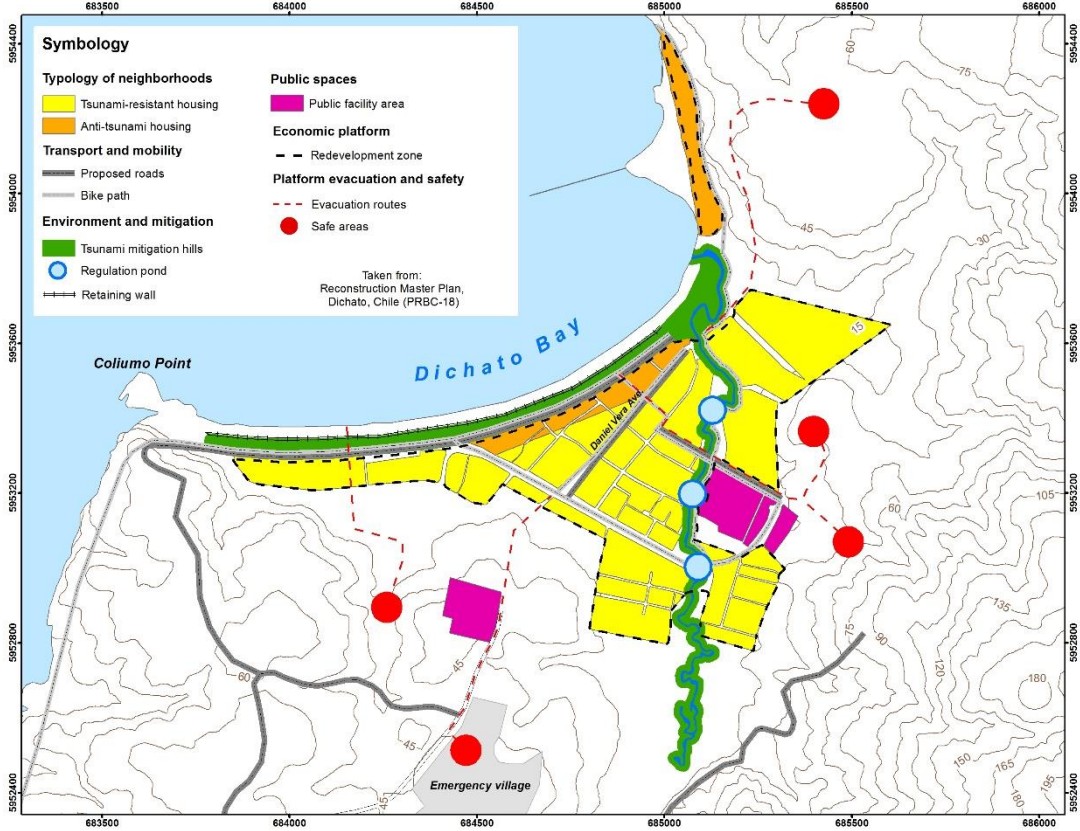


Fig. 8.a) Cluster analysis and 8.b) principal components analysis for different dimensions of
vulnerability by neighborhood. PHY-V (Physical vulnerability), SECO-V (Socio-economic
Vulnerability), EDU-V (Educational vulnerability).

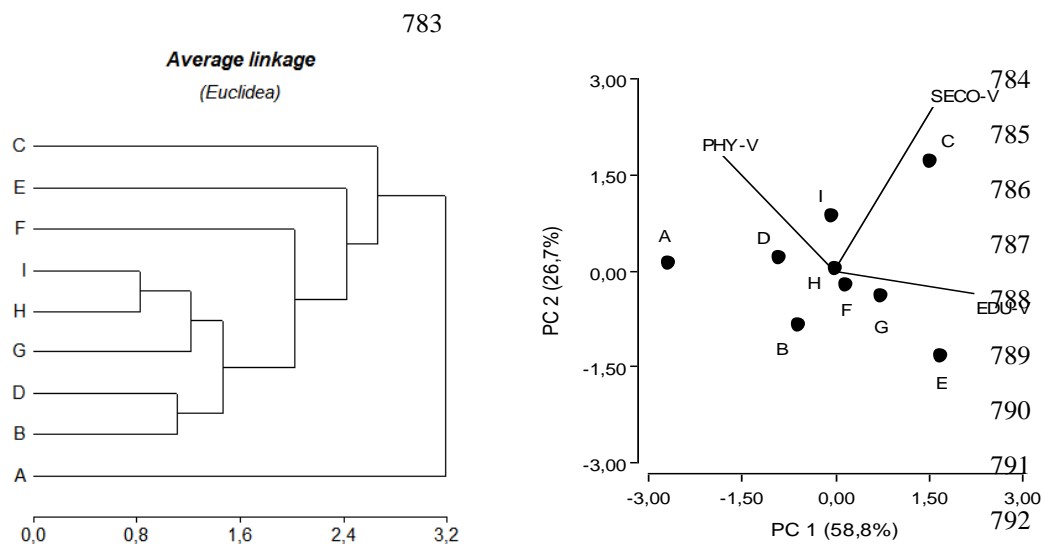



Fig.9. Pre- and post-tsunami vulnerability, Dichato


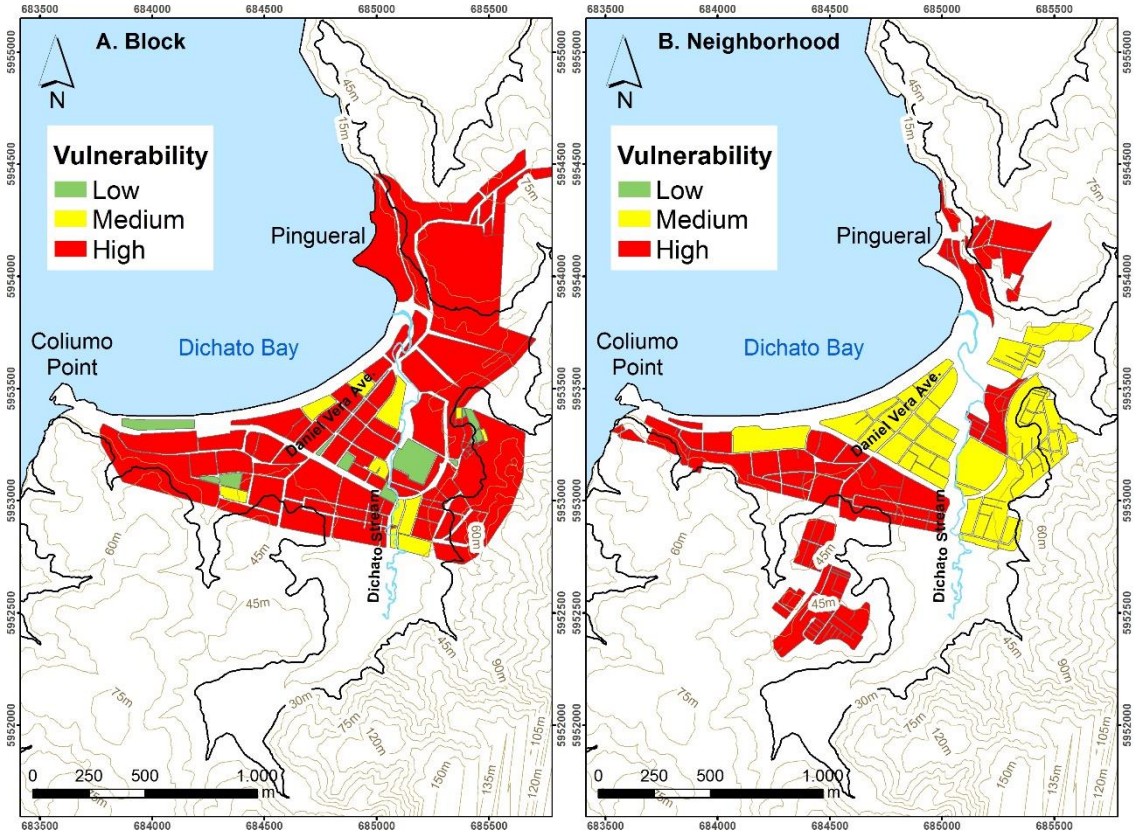

Fig. 10. Tsunami risk areas for pre-event conditions


Fig. 11. Pre-disaster (A, B, C and D) and post-disaster (E and F) housing in Dichato.

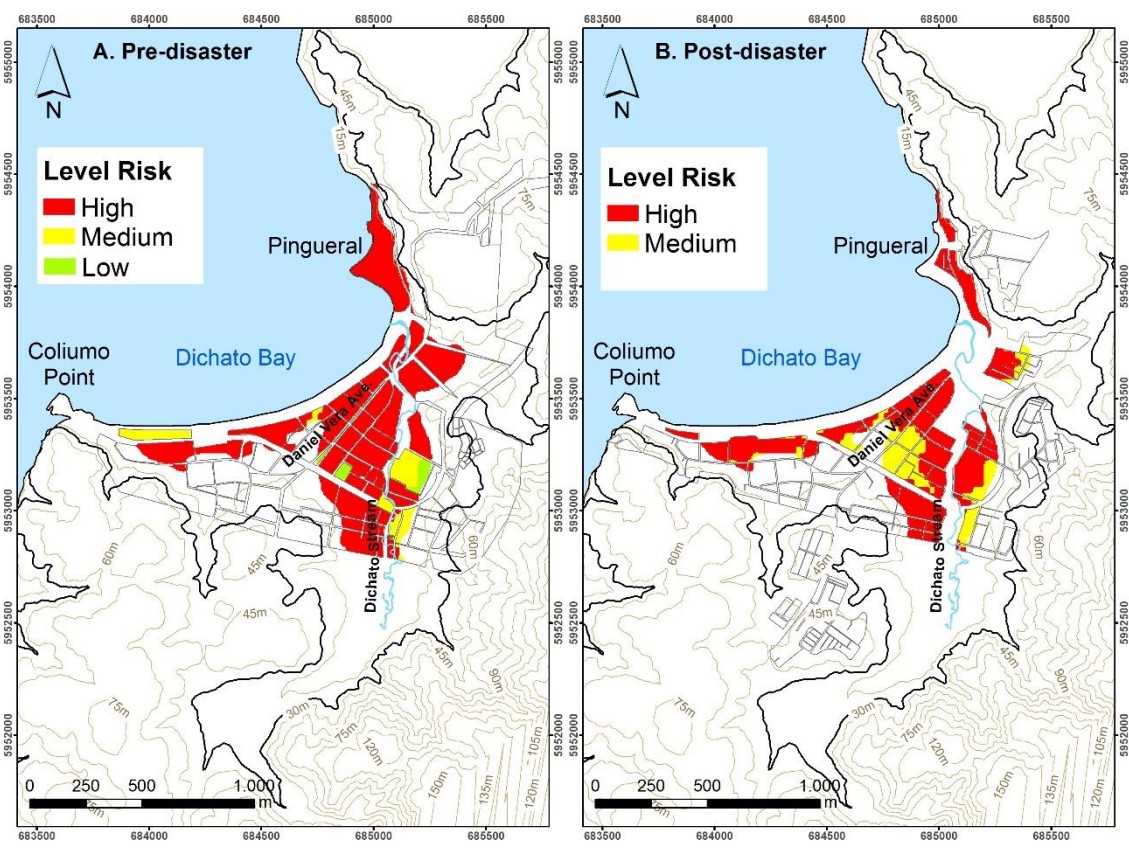

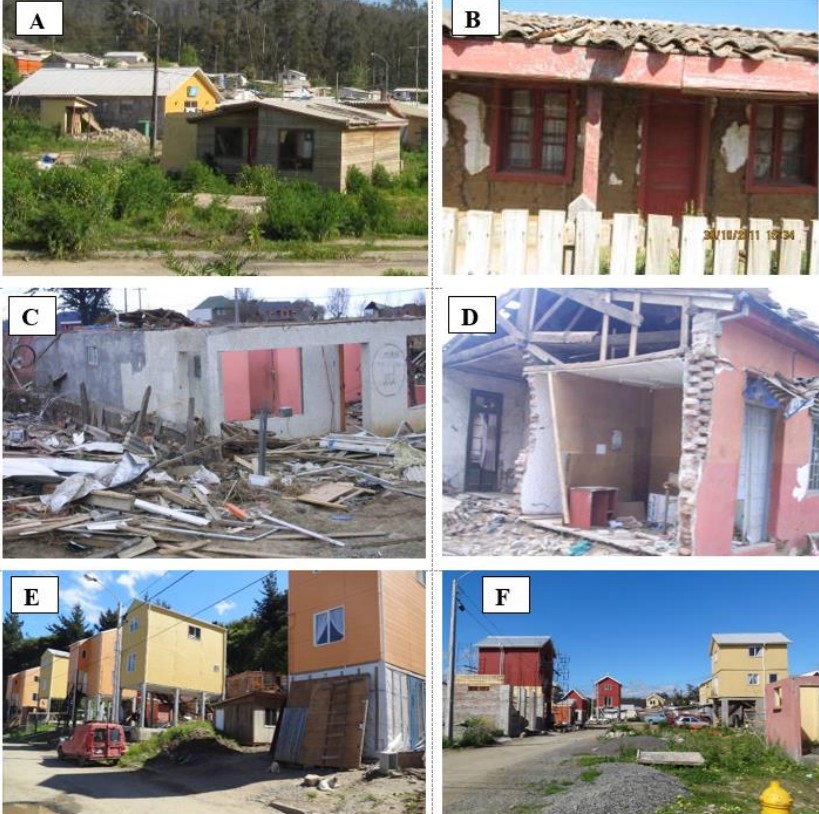

Table 1. Hazard level according to flow depth

| Inundation depth | | |
|---|---|---|
| Range | Description | Hazard Level |
| 0 -.5 m | Knee-high or lower | Low |
| .5 – 2 m | Knee-high to head-high | Medium |
| > 2 m | More than head-high | High |

Reference: modified after Walsh et al. (2005).

Table 2. Hazard level according to flow current velocity

| Current velocity | | |
|---|---|---|
| Range | Descriptor | Hazard Level |
| .1 – 1.35 | Very low and low hazard (speed at which it would be difficult to stand) | Low |
| 1.35 – 2 m/s | Hazard for most | Medium |
| > 2 m/s | Hazard for all, > 5.0 m/s very hazardous | High |

Reference: modified after Jalínek et al. 2012 and González-Riancho et al. 2013

Table 3. Variables associated with each dimension of vulnerability pre- and post-2010 tsunami.

| Physical dimension | | | Socio-economic dimension | | | Educational dimension | | |
|---|---|---|---|---|---|---|---|---|
| Variable | Pre | Post | Variable | Pre | Post | Variable | Pre | Post |
| Housing type | X | X | Population density | X | | Level of knowledge of the phenomenon | X | X |
| Housing material | X | X | Overcrowding level | X | | Knowledge of tsunami warning systems | X | |
| Number of houses | X | | Socio-economic welfare of households (IBS) | X | | Reaction to the event | X | |
| Number of floors | | X | Education level | | X | Knowledge of evacuation routes | | X |
| | | | Labor activity | | X | Knowledge of safe zones | | X |
| | | | Per capita income | | X | Participation in educational programs or lectures | | X |

Table 4. Inundation by tsunami risk matrix, town of Dichato

| X | | Hazard | | |
|---|---|---|---|---|
| Vulnerability | Level | Low    (1) | Medium    (2) | High    (3) |
| | Low    (1) | L    1 X 1 =1 | L    1 X 2 =1 | M 1 X 3 =3 |
| | Medium    (2) | L    2 X 1 =2 | M    2 X 2 =4 | H    2 X 3=6 |
| | High    (3) | M    3 X 1 =3 | H    3 X 2 =6 | H    3 X 3 =9 |

 Risk ranges: Low (1-2), Medium (3-4), High (6-9)

 Table 5. Results with significant analyzed variable differences by neighborhood. Indications are as
 follows: T_S (Total sample), Stat (statistic), df (degrees of freedom), p (p-value), FAS (Fascination) BE-
 AW (being away), COMP (Compatibility), PHY-V (Physical vulnerability), SECO-V (Socio-economic
 vulnerability) EDU-V (Educative vulnerability). Standard deviation in parentheses.

| | T_S | A | B | C | D | E | F | G | H | I | Est | df | p |
|---|---|---|---|---|---|---|---|---|---|---|---|---|---|
| N | 1710 | 155 | 258 | 24 | 325 | 163 | 142 | 66 | 382 | 195 | | | |
| (n) | (172) | (12) | (21) | (17) | (28) | (19) | (19) | (13) | (28) | (15) | | | |
| % of respondents | 10% | 8% | 8% | 71% | 9% | 12% | 13% | 20% | 7% | 8% | | | |
| Anxiety | 30% | 67% | 43% | 24% | 29% | 37% | 32% | 31% | 14% | 7% | 17.440[a] | 8 | .0258 |
| Desires location change (yes) | 36% | 42% | 43% | 18% | 32% | 37% | 58% | 54% | 39% | 0% | 17.577[a] | 8 | .0246 |
| BE-AW | 5.6 (1.3) | 4.7 | 5.4 | 6.5 | 5.7 | 5.8 | 5.7 | 5.1 | 4.9 | 6.1 | 3.58 | | .0007 |
| FAS | 4.4 (1.7) | 3.3 | 4.7 | 4.8 | 4.8 | 4.9 | 3.2 | 4.6 | 4.1 | 5.2 | 3.38 | | .0013 |
| COMP | 5.6 (1.2) | 5.2 | 5.1 | 6.5 | 5.5 | 5.5 | 5.7 | 5.8 | 5.1 | 6.0 | 3.34 | | 0.0015 |
| SECO-V | 6.7 (1.6) | 6.1 | 6.2 | 7.8 | 6.6 | 6.5 | 6.9 | 6.7 | 6.6 | 6.9 | 1.45 | | .1808 |
| PHY-V | 8.8 (2.0) | 11.0 | 8.6 | 8.8 | 9.5 | 6.9 | 7.4 | 8.2 | 9.2 | 9.9 | 51.2 | | <.0001 |
| EDU-V | 7.0 (0.8) | 6.6 | 7.0 | 7.2 | 6.8 | 7.5 | 6.7 | 7.2 | 7.1 | 7.1 | 17.9 | | .0221 |