# Peer review of "1. Introduction"

_Natural Hazards and Earth System Sciences, 2016_

## Referee Comment (RC2)

| 1  | Risk Factors and Perceived Restoration in a Town Destroyed by the 2010 Chile Tsunami                                                                              |
|----|-------------------------------------------------------------------------------------------------------------------------------------------------------------------|
| 2  |                                                                                                                                                                   |
| 3  | Carolina Martínez 1, 6 , Octavio Rojas 2 , Paula Villagra 3 , Rafael Aránguiz 4, 6 , Katia Sáez-Carrillo 5 |
| 4  |                                                                                                                                                                   |
| 5  |                                                                                                                                                                   |
| 6  | 1 Instituto de Geografía, Pontificia Universidad Católica de Chile, Santiago, 8320000, Chile                                                           |
| 7  | 2 Departamento de Planificación Territorial, Universidad de Concepción, Concepción, 4030000, Chile                                                     |
| 8  | 3 Instituto de Ciencias Ambientales y Evolutivas, Universidad Austral de Chile, Valdivia, 5090000, Chile                                               |
| 9  | 4 Departamento de Ingeniería Civil, Universidad Católica de la Santísima Concepción, Concepción,                                                       |
| 10 | 4030000, Chile                                                                                                                                                    |
| 11 | 5 Departamento de Estadística, Universidad de Concepción, Concepción, 4030000, Chile                                                                   |
| 12 | 6 National Research Center for Integrated Natural Disaster Management (CIGIDEN), Santiago, 8320000,                                                    |
| 13 | Chile                                                                                                                                                             |
| 14 |                                                                                                                                                                   |
| 15 | Correspondence to: Carolina Martínez (camartinezr@uc.cl)                                                                                                          |
| 16 |                                                                                                                                                                   |
| 17 |                                                                                                                                                                   |

rs

**ABSTRACT 18**

- 19
- 20
- A large earthquake and ts took place in February 2010, affecting a significant part of the Chilean coast (Maule carthquake (28,8)). Dichato (37° S), a small town located on Coliumo Barras one of 21 s one of
- the most devastated coastal places and is currently under reconstruction. Therefore, the risk 22
- which explain the disaster at that time as well as perceived restoration 6 years after the event were 23
- analyzed in the present paper. Numerical modeling of the 2010 Chile tsunami with four nested grids was 24
- applied to estimate the hazard. Physical, socio-economic and educational dimensions of vulnerability 25
- were analyzed for pre- and post-disaster conditions. A perceived restoration study was performed to 26
- assess the effects of reconstruction on the community and a principal component analysis was applied 27
- for post-disaster conditions. 28
- 29 The vulnerability factors that best explained the extent of the disaster were housing conditions, low
- household incomes and limited knowledge about tsunami events, which conditioned inadequate 30
- reactions to the emergency. These factors still constitute the same risks as a result of the reconstruction 31
- process, establishing that the occurrence of a similar event would result in a similar degree of disaster. 32
- For post-earthquake conditions, it was determined that all neighborhoods have the potential to be 33
- restorative environments soon after a tsunami. However, some neighborhoods are still located in areas 34
- devastated by the 2010 tsunami and present a high vulnerability to future tsunamis. Therefore, it may be 35
- stated that these areas will probably be destroyed again in case of future events. 36
- 37
- Keywords: tsunami, natural risk, territorial planning, social resilience
- 38 39
- 40
- 41

42 **1. Introduction**

43

44 A tsunami is a phenomenon known for its great destructive power in a short period of time; however, the 45 process of post-disaster reconstruction usually lasts a long time and generates significant socio-territorial 46 transformations. A total of seven destructive tsunamis affected the coasts of Indonesia, Samoa, Chile and Japan in only the last decade: 2006, 2007, 2009, 2010 (Feb 27th and Oct 24th) and 2011. These tsunamis 47 took the lives of 237,981 people and generation an estimated US \$456 million in economic losses 48 (Løvholt et al., 2012; Løvholt 2014 et al.) Løvholt et al., 2014 et al.) 49 factors, such as ineffective early warning systems, inadequate management of information by the 50 population, lack of coordination of emergency mechanisms and high levels of social vulnerability (Rofi 51 et al., 2006; Løvholt et al., 2014). Although scientific research has led to significant advances in the 52 generation and propagation mechanisms of these phenomena (Aránguiz et al., 2013; Løvholt et al., 53 2014), other aspects linked to social components (vulnerability and resilience) are less understood, 54 primarily for post-disaster conditions, given social system dynamics and complexity. The latest events 55 have shown that increased mortality may be associated with intrinsic aspects of vulnerability, which in 56 the natural disaster context is defined as the inability of society tq pnd to an event, in this cas 57 dangerous natural phenomenon (Anderson and Woodrow, 1989 i dona, 2001; Wilches-Chau 58 (1993). Intrinsic aspects include population characteristics such as age and gender (Rofi et al., 2006), 59 income levels and job occupations (Birkman 2007), ideological and cultural factors 60 knowledge and inadequate reactions to the emergency (Ruan and Hogben, 2007). Others, 61 gh a line of still incipient work, have established that factors associated with social capital and territorial identity 62 foster social resilience, which would be an enabling framework to overcome the negative effects of a 63 disturbance (Pelling, 2003). 64 The 2010 Chile tsunami showed the high fragility of social and institutional systems in created areas, as 65 et al., 2010; by المسنع et al., 2010; by المسنع et al., 66 significant destruction alo 2011; Contreras et al., 201 amillo 
[revised manuscript text omitted]
 and environmental restoration                                                                                                                                                                                                                                                                                                                                                                                                                                                                                                                                                                                                                                                                                                                                                                                                                                                                                                                                                                                                                                                                                                                                                                                                                                                                                                                                                                                                                                                                                                                                                                                                                                                                                                                                                                                                                                                                                                                                                                                                                                                                               |  |  |
| 151 | The surface of a stability which for stars determined data a birry different base of the surface of the stars and a                                                                                                                                                                                                                                                                                                                                                                                                                                                                                                                                                                                                                                                                                                                                                                                                                                                                                                                                                                                                                                                                                                                                                                                                                                                                                                                                                                                                                                                                                                                                                                                                                                                                                                                                                                                                                                                                                                                                                                                                           |  |  |
| 152 | In order to establish which factors determined the achieved hazard level as well as the effects generated                                                                                                                                                                                                                                                                                                                                                                                                                                                                                                                                                                                                                                                                                                                                                                                                                                                                                                                                                                                                                                                                                                                                                                                                                                                                                                                                                                                                                                                                                                                                                                                                                                                                                                                                                                                                                                                                                                                                                                                                                     |  |  |
| 153 | by the post-disaster reconstruction process in shaping new risk areas, the vulnerability analysis was                                                                                                                                                                                                                                                                                                                                                                                                                                                                                                                                                                                                                                                                                                                                                                                                                                                                                                                                                                                                                                                                                                                                                                                                                                                                                                                                                                                                                                                                                                                                                                                                                                                                                                                                                                                                                                                                                                                                                                                                                         |  |  |
| 154 | For total vulnerability analysis, variables selected for both scenarios were representative of physical                                                                                                                                                                                                                                                                                                                                                                                                                                                                                                                                                                                                                                                                                                                                                                                                                                                                                                                                                                                                                                                                                                                                                                                                                                                                                                                                                                                                                                                                                                                                                                                                                                                                                                                                                                                                                                                                                                                                                                                                                       |  |  |
| 155 | socio-economic and educational dimensions: however, some variables were modified according to                                                                                                                                                                                                                                                                                                                                                                                                                                                                                                                                                                                                                                                                                                                                                                                                                                                                                                                                                                                                                                                                                                                                                                                                                                                                                                                                                                                                                                                                                                                                                                                                                                                                                                                                                                                                                                                                                                                                                                                                                                 |  |  |
| 150 | socio-economic and educational dimensions; however, some variables were modified according to                                                                                                                                                                                                                                                                                                                                                                                                                                                                                                                                                                                                                                                                                                                                                                                                                                                                                                                                                                                                                                                                                                                                                                                                                                                                                                                                                                                                                                                                                                                                                                                                                                                                                                                                                                                                                                                                                                                                                                                                                                 |  |  |
| 158 | pre/post-disaster conditions (Table 3). In the case of pre-disaster combines, the analysis unit corresponded                                                                                                                                                                                                                                                                                                                                                                                                                                                                                                                                                                                                                                                                                                                                                                                                                                                                                                                                                                                                                                                                                                                                                                                                                                                                                                                                                                                                                                                                                                                                                                                                                                                                                                                                                                                                                                                                                                                                                                                                                  |  |  |
| 150 | the analysis unit was the neighborhood, which due to the destruction caused by the tsunami and the                                                                                                                                                                                                                                                                                                                                                                                                                                                                                                                                                                                                                                                                                                                                                                                                                                                                                                                                                                                                                                                                                                                                                                                                                                                                                                                                                                                                                                                                                                                                                                                                                                                                                                                                                                                                                                                                                                                                                                                                                            |  |  |
| 160 | absence of census data, was defined according to similarities of the post-disaster buildings (Fig. 4).                                                                                                                                                                                                                                                                                                                                                                                                                                                                                                                                                                                                                                                                                                                                                                                                                                                                                                                                                                                                                                                                                                                                                                                                                                                                                                                                                                                                                                                                                                                                                                                                                                                                                                                                                                                                                                                                                                                                                                                                                        |  |  |
| 161 | Variables were incorporated into the GIS ArcGis 10.1 to generate thematic maps and synthesis charts                                                                                                                                                                                                                                                                                                                                                                                                                                                                                                                                                                                                                                                                                                                                                                                                                                                                                                                                                                                                                                                                                                                                                                                                                                                                                                                                                                                                                                                                                                                                                                                                                                                                                                                                                                                                                                                                                                                                                                                                                           |  |  |
| 162 | through map algebra.                                                                                                                                                                                                                                                                                                                                                                                                                                                                                                                                                                                                                                                                                                                                                                                                                                                                                                                                                                                                                                                                                                                                                                                                                                                                                                                                                                                                                                                                                                                                                                                                                                                                                                                                                                                                                                                                                                                                                                                                                                                                                                          |  |  |
| 163 |                                                                                                                                                                                                                                                                                                                                                                                                                                                                                                                                                                                                                                                                                                                                                                                                                                                                                                                                                                                                                                                                                                                                                                                                                                                                                                                                                                                                                                                                                                                                                                                                                                                                                                                                                                                                                                                                                                                                                                                                                                                                                                                               |  |  |
| 164 | The capacity of the neighborhoods of Dichato to provide restorative experiences post-disaster was                                                                                                                                                                                                                                                                                                                                                                                                                                                                                                                                                                                                                                                                                                                                                                                                                                                                                                                                                                                                                                                                                                                                                                                                                                                                                                                                                                                                                                                                                                                                                                                                                                                                                                                                                                                                                                                                                                                                                                                                                             |  |  |
| 165 | assessed through a perceived restoration study (Hartig et al., 1997). The inhabitants assessed their                                                                                                                                                                                                                                                                                                                                                                                                                                                                                                                                                                                                                                                                                                                                                                                                                                                                                                                                                                                                                                                                                                                                                                                                                                                                                                                                                                                                                                                                                                                                                                                                                                                                                                                                                                                                                                                                                                                                                                                                                          |  |  |
| 166 | neighborhoods by means of the Perceived Restorative Scale (PRS), an instrument constructed based on                                                                                                                                                                                                                                                                                                                                                                                                                                                                                                                                                                                                                                                                                                                                                                                                                                                                                                                                                                                                                                                                                                                                                                                                                                                                                                                                                                                                                                                                                                                                                                                                                                                                                                                                                                                                                                                                                                                                                                                                                           |  |  |
| 167 | the Attention Restoration Theory (Kaplan and Kaplan, 1989). The neighborhoods were defined as units of                                                                                                                                                                                                                                                                                                                                                                                                                                                                                                                                                                                                                                                                                                                                                                                                                                                                                                                                                                                                                                                                                                                                                                                                                                                                                                                                                                                                                                                                                                                                                                                                                                                                                                                                                                                                                                                                                                                                                                                                                        |  |  |
| 168 | study (Fig. 4). The PRS has been used to identify landscape attributes that can be restorative to people                                                                                                                                                                                                                                                                                                                                                                                                                                                                                                                                                                                                                                                                                                                                                                                                                                                                                                                                                                                                                                                                                                                                                                                                                                                                                                                                                                                                                                                                                                                                                                                                                                                                                                                                                                                                                                                                                                                                                                                                                      |  |  |
| 169 | subjected to high levels of stress and mental fatigue (Hartig et al., 1997; Korpela and Hartig, 1996; Ulrich                                                                                                                                                                                                                                                                                                                                                                                                                                                                                                                                                                                                                                                                                                                                                                                                                                                                                                                                                                                                                                                                                                                                                                                                                                                                                                                                                                                                                                                                                                                                                                                                                                                                                                                                                                                                                                                                                                                                                                                                                  |  |  |
| 170 | et al., 1991). Access to restorative environments is also crucial in cities prone to natural disasters, such as                                                                                                                                                                                                                                                                                                                                                                                                                                                                                                                                                                                                                                                                                                                                                                                                                                                                                                                                                                                                                                                                                                                                                                                                                                                                                                                                                                                                                                                                                                                                                                                                                                                                                                                                                                                                                                                                                                                                                                                                               |  |  |
| 171 | tsunamis. Three factors were used to evaluate the interaction of people with the neighborhood they inhabit:                                                                                                                                                                                                                                                                                                                                                                                                                                                                                                                                                                                                                                                                                                                                                                                                                                                                                                                                                                                                                                                                                                                                                                                                                                                                                                                                                                                                                                                                                                                                                                                                                                                                                                                                                                                                                                                                                                                                                                                                                   |  |  |
| 172 | being away (BE-AW), which reflects the need to escape from everyday life or daily mental activities that                                                                                                                                                                                                                                                                                                                                                                                                                                                                                                                                                                                                                                                                                                                                                                                                                                                                                                                                                                                                                                                                                                                                                                                                                                                                                                                                                                                                                                                                                                                                                                                                                                                                                                                                                                                                                                                                                                                                                                                                                      |  |  |
| 173 | require major concentration; fascination (FAS), which is found in environments that attract and hold our                                                                                                                                                                                                                                                                                                                                                                                                                                                                                                                                                                                                                                                                                                                                                                                                                                                                                                                                                                                                                                                                                                                                                                                                                                                                                                                                                                                                                                                                                                                                                                                                                                                                                                                                                                                                                                                                                                                                                                                                                      |  |  |
| 174 | attention without any effort; and compatibility (COMP), which refers to a sense of oneness with                                                                                                                                                                                                                                                                                                                                                                                                                                                                                                                                                                                                                                                                                                                                                                                                                                                                                                                                                                                                                                                                                                                                                                                                                                                                                                                                                                                                                                                                                                                                                                                                                                                                                                                                                                                                                                                                                                                                                                                                                               |  |  |
| 175 | environments that provides the capability to meet our desires and needs. Each factor was evaluated using                                                                                                                                                                                                                                                                                                                                                                                                                                                                                                                                                                                                                                                                                                                                                                                                                                                                                                                                                                                                                                                                                                                                                                                                                                                                                                                                                                                                                                                                                                                                                                                                                                                                                                                                                                                                                                                                                                                                                                                                                      |  |  |

---

## Referee Comment (RC1) · Anonymous Referee #1 · 12 Oct 2016

General comments The authors analyze in this paper tsunami inundation risks pre- and post-disaster in one of the coastal towns most affected by the earthquake and tsunami on Feb. 27, 2010, which presented an intense transformation as a result of post-disaster reconstruction. They aim at understanding whether this reconstruction process has reduced vulnerability and provided a restorative urban system, which enhance urban resilience, or if it has generated new risk areas.

The topic is of relevance for this journal and for the international scientific community since reconstruction processes are usually not assessed and/or compared with the situation before the event. This information could be relevant to optimize reconstruction and to avoid repeating the same mistakes in urban planning.

The paper is well structured. However, some minor changes are required.

Specific comments

The abstract needs additional work since (1) it is lacking the objective of the paper. Without the objective, the listing of technical tasks carried out does not give the reader an overall understanding of the work; (2) the description of the perceived restoration study, which is not a common and widely known topic, is not clear; (3) there is a confusing use of terms, i.e. "risk factors", "vulnerability factors", "these factors constitute the same risks.". The authors should be consistent along the paper with chosen terminology; (4) the statement "these areas will probably be destroyed again" is too assertive, considering the existing uncertainties other type of expression may be more adequate.

Methodology - Vulnerability and risk assessment:

The vulnerability pre- and post-tsunami variables associated with each dimension could be cited in the text. Besides, the authors should justify why some variables were modified according to pre/post-disaster conditions. Why are the authors not using the same variables? Is it due to lack of data? Scientific approach? Are pre/post-disaster conditions comparable measuring different variables? The authors should clarify whether this decision affects or not the final results.

It is not clear if (and how) the vulnerability assessment combines the vulnerability variables and the perceived restoration study or not. Therefore, it is not clear as well if both analyses feed the risk matrix or not.

In order to facilitate readers from different disciplines understand the analysis, it should be better explained why the chosen statistical methods are applied. For example, what are the benefits of clustering against other options?

The results provided are not fully understandable. The description of the type of result and the percentages are confusing. Better explanations of the results would help the reader to better follow the line of argumentation:

Vulnerability pre-disaster

- High V: 51% of census blocks = 47% of inundated area = 57% of total population -
Average V: 73% of households = 61% of inundated area = 67% of total population

Post-event conditions:

- Affected: 72% of census blocks = 70% of housing = 73% of total population

Vulnerability post-disaster: analysis of neighbourhoods and restoration values

Secondly, the analysis of neighbourhoods presents the clustering which, although useful, is not well justified, neither in the methodology section (why was this method selected? what is it expected to provide?), nor in the results section (what is the relevance of these results besides the fact of grouping neighbourhoods?). Additional explanations dealing with the relevance of the results should be provided.

The Conclusions section should also provide some remarks about the contributions of the proposed method.

Technical corrections

P3, line 52. Maybe some words missing, suggestion in brackets: "Although scientific research has led to significant advances in [the understanding of] the generation and propagation mechanisms of these phenomena",

P3, line 81. "...in Chile, however, physical and social dimensions are the least considered in post-disaster planning." However, in p11, line 403 it is said that "This situation is explained by the emphasis on physical rather than social reconstruction..." Do you maybe mean, in line 81, psychological and social dimensions?

P5, line 152. "In order to establish which factors determined the achieved hazard level...". According to literature on this topic, the terminology of this sentence is confusing. The vulnerability factors may influence the impacts, but not the hazard level. Please justify.

Tables need reordering, there are two Table 3.

---

## Author Comment (AC1) · 24 Feb 2017

Abstract: 1. Include the objective of the paper in the abstract.

Answer: It was incorporated into the following sentence: Therefore, the objective of this research is to analyze the risk factors that explain the disaster at that time as well as perceived restoration 6 years after the event.

2. The description of the perceived restoration study, which is not a common and widely known topic, is not clear.

Answer: A perceived restoration study was performed to assess the effects of reconstruction on the community. It consisted of evaluating the capacity of the new neighborhoods to provide restorative experiences in case of disaster by asking community

members to assess 15 items associated with the Being Away, Fascination and Compatibility factors found in the Perceived Restorative Scale. A 1-7 Likert scale was used during the evaluation.

3. Separate the use of "factor" from references to vulnerability.

Answer: It was decided to use this term to refer to risk. "Variable" was used to refer to vulnerability.

4. Use of the idea that "these areas will probably be destroyed again."

Answer: It was decided to eliminate this sentence.

Methodology - Vulnerability and risk assessment:

1. The vulnerability pre- and post-tsunami variables associated with each dimension could be cited in the text.

Answer: This was not deemed necessary since the variables are indicated and detailed in Table 3.

2. The authors should justify why some variables were modified according to pre/post-disaster conditions.

Answer: This justification appears in lines 170 and 180. Some of the variables used in pre-disaster conditions were from the 2002 census; however, census data regarding was unavailable for the same variables for post-disaster conditions. Therefore, it was decided to select representative variables for each dimension of vulnerability. In Chile, the next census is planned for 2017.

3. The authors should justify why some variables were modified according to pre/post-disaster conditions. Why are the authors not using the same variables? Is it due to lack of data? Scientific approach? Are pre/post-disaster conditions comparable measuring different variables? The authors should clarify whether this decision affects or not the final results.

Answer: Due to the lack of post-disaster census data, some variables were not similar for pre- and post-disaster conditions for the same analysis unit. It was attempted to overcome this difficulty by incorporating representative variables for each dimension of vulnerability. This aspect was added to the Discussion section, between lines 441 and 446.

4. It is not clear if (and how) the vulnerability assessment combines the vulnerability variables and the perceived restoration study or not. Therefore, it is not clear as well if both analyses feed the risk matrix or not.

Answer: The vulnerability variables were given equal weight in the final matrix since prior studies in the area that included the dimension of vulnerability in risk such as Martínez et al., (2012) and Rojas et al., (2014) used similar criteria, which have proved representative of local conditions. Both studies (vulnerability and perceived restoration) were complementary, but restoration was not included in the vulnerability matrix. The risk matrix considers only risk factors (threat and vulnerability), in accordance with Blakie et al., (1994), and therefore restoration was included only in the principal component and cluster analyses for post-disaster conditions. To improve the presentation of these results, a section was created for the perceived restoration study (line 340).

5. In order to facilitate readers from different disciplines understand the analysis, it should be better explained why the chosen statistical methods are applied. For example, what are the benefits of clustering against other options?

Answer: There are various advantages to using multivariate methods in this type of investigation. Cluster analysis allows similarities and differences between various neighborhoods to be easily observed. Principal component analysis allows better observation of the association of the various analyzed variables with each particular analysis unit.

6. The results provided are not fully understandable. The description of the type of result and the percentages are confusing. Better explanations of the results would

help the reader to better follow the line of argumentation:

Vulnerability pre-disaster:

- High V: 51% of census blocks = 47% of inundated area = 57% of total population - Average V: 73% of households = 61% of inundated area = 67% of total population

Post-event conditions: - Affected: 72% of census blocks = 70% of housing = 73% of total population

Answer: A new paragraph was written to improve this aspect, between lines 372 and 376.

7. Vulnerability post-disaster: analysis of neighbourhoods and restoration values Secondly, the analysis of neighbourhoods presents the clustering which, although useful, is not well justified, neither in the methodology section (why was this method selected? what is it expected to provide?), nor in the results section (what is the relevance of these results besides the fact of grouping neighbourhoods?). Additional explanations dealing with the relevance of the results should be provided.

Answer: This analysis was eliminated from the study. The relevant information for this paper is provided by the descriptive statistical analysis that was performed. It provides information on which neighborhoods are more restorative than others (the old or new neighborhoods) and which elements of the neighborhoods contribute to a restorative experience (natural or built features). This was also clarified in section 3.3.

8. The Conclusions section should also provide some remarks about the contributions of the proposed method.

Answer: A sentence was incorporated into the Conclusions section, between lines 519 and 523.

Technical corrections

P3, line 52. Maybe some words missing, suggestion in brackets: "Although scientific

research has led to significant advances in [the understanding of] the generation and propagation mechanisms of these phenomena",

Answer: The sentence was corrected in accordance with the referee's proposal.

P3, line 81. "...in Chile, however, physical and social dimensions are the least considered in post-disaster planning." However, in p11, line 403 it is said that "This situation is explained by the emphasis on physical rather than social reconstruction..." Do you maybe mean, in line 81, psychological and social dimensions?

Answer: The sentence was corrected in accordance with the referee's proposal.

P5, line 152. "In order to establish which factors determined the achieved hazard level...". According to literature on this topic, the terminology of this sentence is confusing. The vulnerability factors may influence the impacts, but not the hazard level. Please justify.

Answer: In effect, there was a writing problem here. The text was corrected to the following:

Tables need reordering, there are two Table 3.

Answer: Table 3 was reordered.

Please also note the supplement to this comment:
http://www.nat-hazards-earth-syst-sci-discuss.net/nhess-2016-256/nhess-2016-256-AC1-supplement.pdf

---

## Author Comment (AC2) · 24 Feb 2017

1. I think the paper would earn in readability and would attract a larger audience if section 4 can be made more "loquacious", instead of being a list of percentages.

Answer: This section was rewritten, with the comments of Referee 1 incorporated as well, in order to improve its readability for a larger audience.

2. I recommend the authors to carefully check the references. Many of those cited in the text are missing in the references list at the end of the paper.

Answer: The missing references were added and those used in the text were checked.

3. The authors thank the referee (A. Armigliato) for the comments on the manuscript, which were incorporated into the text.

[Figure]

P.S. The corrected version of the manuscript is attached.

Please also note the supplement to this comment:
http://www.nat-hazards-earth-syst-sci-discuss.net/nhess-2016-256/nhess-2016-256-AC2-supplement.pdf